palaeontology

coprolites, palaeoecology, Dinosauromorpha, beetle elytra, insectivore

**Author for correspondence:**
Grzegorz Niedźwiedzki
e-mail: grzegorz.niedzwiedzki@ebc.uu.se

# Beetle-bearing coprolites possibly reveal the diet of a Late Triassic dinosauriform

Martin Qvarnström[1], Joel Vikberg Wernström[1], Rafał Piechowski[2,3], Mateusz Tałanda[3], Per E. Ahlberg[1] and Grzegorz Niedźwiedzki[1]

[1]Department of Organismal Biology, Evolutionary Biology Centre, Uppsala University, Norbyvägen 18A, 752 36 Uppsala, Sweden
[2]Institute of Paleobiology, Polish Academy of Sciences, Twarda 51/55, 00-818 Warszawa, Poland
[3]Department of Palaeobiology and Evolution, Faculty of Biology, Biological and Chemical Research Centre, University of Warsaw, Żwirki i Wigury 101, 02-089 Warszawa, Poland

MQ, 0000-0001-7998-2243; MT, 0000-0003-3358-9539; GN, 0000-0002-4775-5254

Diets of extinct animals can be difficult to analyse if no direct evidence, such as gut contents, is preserved in association with body fossils. Inclusions from coprolites (fossil faeces), however, may also reflect the diet of the host animal and become especially informative if the coprolite producer link can be established. Here we describe, based on propagation phase-contrast synchrotron microtomography (PPC-SRμCT), the contents of five morphologically similar coprolites collected from two fossil-bearing intervals from the highly fossiliferous Upper Triassic locality at Krasiejów in Silesia, Poland. Beetle remains, mostly elytra, and unidentified exoskeleton fragments of arthropods are the most conspicuous inclusions found in the coprolites. The abundance of these inclusions suggests that the coprolite producer deliberately targeted beetles and similar small terrestrial invertebrates as prey, but the relatively large size of the coprolites shows that it was not itself a small animal. The best candidate from the body fossil record of the locality is the dinosauriform *Silesaurus opolensis* Dzik, 2003, which had an anatomy in several ways similar to those of bird-like neotheropod dinosaurs and modern birds. We hypothesize that the beak-like jaws of *S. opolensis* were used to efficiently peck small insects off the ground, a feeding behaviour analogous to some extant birds.

## 1. Introduction

Vertebrate coprolites are common elements in marine and non-marine fossiliferous deposits from the Palaeozoic to recent. In

the same way as footprints, they serve as important palaeoichnological proxies for the presence and diversity of vertebrates in ancient ecosystems (e.g. [1,2]). Coprolites especially provide unique information about the feeding habits and digestive physiology of the host animals, which in turn gives insights into the different trophic levels of ancient ecosystems [2–5]. Much of this palaeoecological information lies hidden in the enclosed, but often well-preserved, remains which comprise food residues, microbiota and parasites. The high preservation potential of these organic inclusions has been attributed to early lithification of the faeces, a process thought to be facilitated by high phosphate content and bacterial autolithification [6–8]. Consequently, even soft tissues can become fossilized within coprolites [6,7,9]. Inclusions described from coprolites include plant remains, bones, teeth, hairs, feathers, muscle cells, invertebrate exoskeleton, insect wings, parasite bodies and eggs [6,9–13]. However, the inclusions are often hard to study because they have been processed in the digestive system of the host animal; as a result, they are often fragmentary and chaotically organized in the heterogeneous fossilized coprolite matrix.

Linking coprolites to their producer is also challenging (e.g. [7,14,15]), but naturally one of the main goals studying coprolites [5,11,16]. The most valuable tools to solve this challenge lies in the shape, size and contents of the coprolites, as well as their association with tracks or skeletal remains. Large coprolite sizes alone can, for example, allow an attribution of coprolites to big apex predators [4]. Size, and in particular the diameter of the specimens (which has a positive correlation to the body size/weight of the host animal [17–19]), can also be used to statistically separate groups of coprolites. This practice is often used to estimate recent population sizes based on faeces, as well as to separate sympatric and similarly sized scat producers [19]. Moreover, characteristic features related to shape such as the presence or the absence of spirals, segmentation and morphology of the ends are also characters that can be useful to discriminate between coprolite morphotypes. Vertebrates that have been successfully matched to their coprolites include fish [10,20–23], crocodilians [23,24], non-avian dinosaurs [3,4,25], early synapsids [11] and mammals [18,26,27].

Coprolite size and shape are easily studied macroscopically. The inclusions, however, have traditionally been studied by optical microscopy of thin sections or by scanning electron microscopy. These destructive methods provide only a limited and partly random representation of the coprolite contents. Recently, propagation phase-contrast synchrotron microtomography (PPC-SRμCT) was shown to be a powerful technique to analyse the inclusions of the entire coprolites, non-destructively, in three dimensions, and high resolution [28]. Two coprolites from the Late Triassic assemblage in Krasiejów (Silesia, Poland) have already been analysed using this technique [28]. One coprolite was spiral and contained a partly articulated ganoid fish and bivalve remains, suggesting that the coprolite was produced by the lungfish *Ptychoceratodus*. The other contained numerous beetle remains implying an insectivorous animal as the producer. The aim of this study was to analyse all synchrotron-scanned specimens of the same morphotype (and locality) as this beetle-bearing coprolite in order to deduce the producer, and to furthermost extent, its diet and palaeoecology.

# 2. Material and methods

## 2.1. Fossil material

This study used the following specimens in the collections of the Institute of Paleobiology, Polish Academy of Sciences (Warsaw): ZPAL AbIII/3402, 3208-3411 (figure 1).

## 2.2. Phase-contrast synchrotron microtomography

The coprolites were scanned using propagation phase-contrast synchrotron microtomography (PPC-SRμCT) at beamline ID19 of the European Synchrotron Radiation Facility (ESRF) in Grenoble, France, in separate sessions with somewhat different settings. In the cases were the settings differ and nothing else is stated, the first mentioned settings applies for ZPAL AbIII/3409 and ZPAL AbIII/3410, the second for ZPAL AbIII/3402 and ZPAL AbIII/3411, and the third (last) for ZPAL AbIII/3408. The coprolites were scanned in vertical series of 4, 5 and 4 mm, respectively, in the so-called half acquisition mode meaning that the centre at rotation was set at the side of the camera field of view (resulting in a doubling of the reconstructed field of view). The propagation distance, or the distance between the sample on the rotation stage and the camera, was set at 2800 mm. The camera was a sCMOS PCO edge 5.5 detector, mounted on optical devices bringing isotropic voxel sizes of 6.36, 6.54

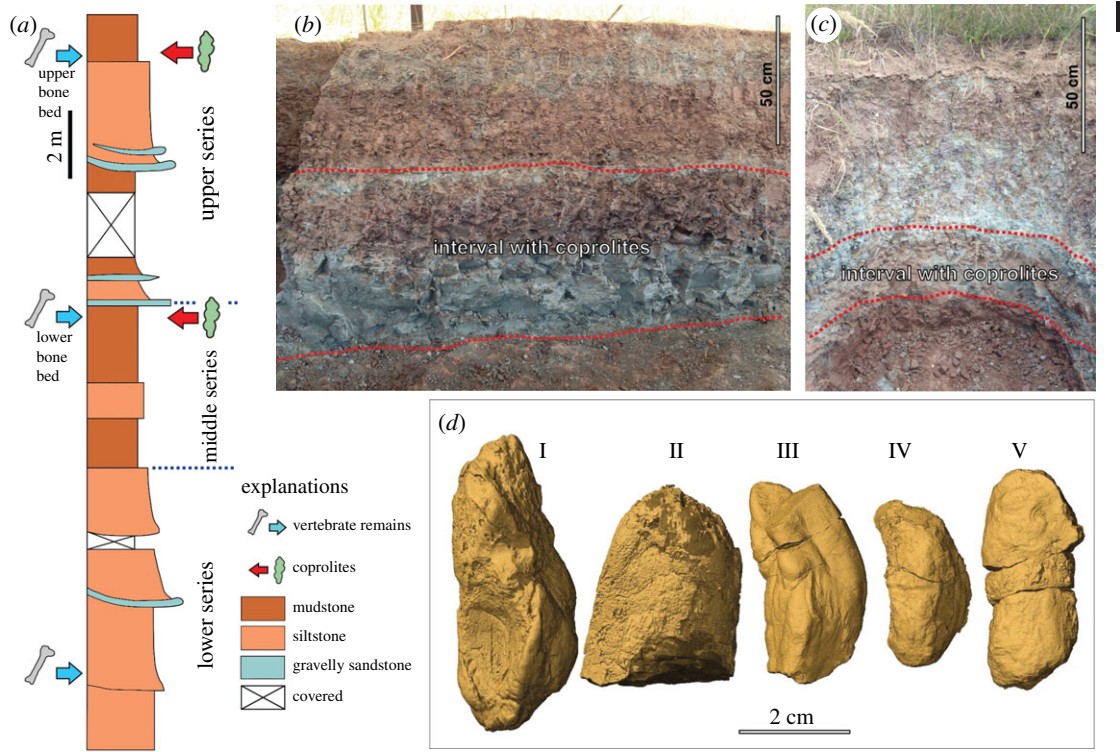

**Figure 1.** Stratigraphic column of the Krasiejów (Silesia, Poland) site. (*a*) Composite lithostratigraphic column compiled from exposures measured in the clay-pit (based on Gruszka & Zieliński [29]) with positions of bone-bearing intervals and layer with coprolites. (*b*,*c*) Photographs of the lower (*b*) and upper (*c*) coprolite-bearing intervals (fieldworks in 2013). (*d*) Three-dimensional surface models of the studied coprolite specimens (I, ZPAL AbIII/3402; II, ZPAL AbIII/3408; III, ZPAL AbIII/3411; IV, ZPAL AbIII/3410; V, ZPAL AbIII/3409).

and 13.4 μm respectively, and coupled to a 1000-μm thick GGG:Eu (gadolinium gallium garnet doped with europium) versus a 500-μm thick LuAG:Ce (lutetium aluminium garnet doped with caesium) scintillator (only ZPAL AbIII/3409 and ZPAL AbIII/3410). The beam was produced by a W150 wiggler (11 dipoles, 150 mm period) with a gap of 51, 48 and 50 mm, respectively, and was filtered with 2.8 mm aluminium (5.6 mm for ZPAL AbIII/3402 and ZPAL AbIII/3411) and 6 mm copper (5 mm for ZPAL AbIII/3402 and ZPAL AbIII/3411). The resulting detected spectrum had average energies of 112, 111 and 113 keV, respectively. Each sub scan was performed using 6000 projections of 0.05 (ZPAL AbIII/3409 and ZPAL AbIII/3410), respectively 0.02 s each over 360°.

# 3. Results and discussion

## 3.1. The locality

The nearly 30 m thick Upper Triassic deposit exposed in Krasiejów comprises two major bone-bearing intervals, each of about 1–1.5 m thickness [30] (figure 1). Both intervals contain large amounts of fossil remains belonging to two ecological communities—a freshwater and a terrestrial community [30]. The freshwater community comprises dipnoans (including the lungfish *Ptychoceratodus roemeri*), small actinopterygians, the temnospondyl *Metoposaurus krasiejovensis*, the large temnospondyl *Cyclotosaurus intermedius,* the large phytosaur *Parasuchus* sp. and a diverse invertebrate fauna. The terrestrial community was composed of small diapsids (e.g. sphenodonts), the gliding archosauromorph *Ozimek volans*, the dinosauriform *Silesaurus opolensis*, the large carnivorous 'rauisuchid' *Polonosuchus silesiacus* and the omnivorous or herbivorous aetosaur *Stagonolepis olenkae*.

The precise age of the assemblage is difficult to determine because of a lack of radiometric dates and diagnostic palynomorphs but plant macrofossils, the vertebrate community and certain invertebrate fossils (e.g. conchostracans) are compatible with a Late Carnian age [30–32].

**Table 1.** Measurements of the coprolites, number of elytra in each coprolite and descriptions of other inclusions. The diameter refers to the maximum diameter. Lengths and diameters are expressed in mm.

| specimen | length | diameter | no. elytra | other inclusions | comments |
|---|---|---|---|---|---|
| ZPAL AbIII/3402 (lower interval) | 54.5 | 22 | >18 | Beetle remains: two tibiae, a carabid prosternum and two pronotums. Abundant exoskeletal fragments. | |
| ZPAL AbIII/3408 (upper interval) | >37.5 | 22 | >15 | An abdomen of an unknown insect, and enigmatics including a curved object with denticles. | The coprolite is in a concretion and is not complete. |
| ZPAL AbIII/3409 (lower interval) | 40 | 19 | 2 | Small fragments of possible insect origin. Most inclusions are small and poorly preserved. | Various internal structures including secondary mineralized voids and a heterogeneous matrix. |
| ZPAL AbIII/3410 (upper interval) | 31 | 16 | >10 | An abdomen of an unknown insect and an apparent exoskeletal plate of an insect thorax. | |
| ZPAL AbIII/3411 (lower interval) | 39 | 19 | 3 | Very abundant small potential insect fragments. | |

## 3.2. Coprolites

The coprolites were collected from the lower and upper bone-bearing intervals at the Krasiejów locality (figure 1). Both fossiliferous intervals are well exposed on the southeastern side of the clay-pit and have been studied in detail from sedimentological and taphonomical aspects [29,30,33]. All five studied coprolites (ZPAL AbIII/3402, 3408–3411; figure 1 and table 1) are complete or near-complete and belong to a morphotype that is characterized by elongated, rounded to slightly flattened specimens, with characteristic irregular surface structures (figure 1). The coprolites have a thin, smooth outer coating and are grey to brown in colour. They range in size from 31 to 54.5 mm in length and from 16 to 22 mm in maximum diameter (table 1).

The coprolites contain abundant beetle remains and other arthropod inclusions that are in most cases too fragmented to be identified. Beetle elytra constitute the most common identifiable remains and are present in all coprolites (figures 2 and 3). Beetle elytra are rare remains in the fossil record of Krasiejów. Until now, only a few and usually fragmentary specimens have been collected from this site [30]. The elytra from coprolites are generally well preserved and complete although some specimens are damaged or sheared from the alimentary tract of the producer, microbial decay and/or diagenesis. Other identifiable inclusions comprise: two beetle tibiae, a carabid prosternum, beetle pronotums, an ostracod, bacterial colonies and two similar abdomens of an unidentified arthropod (figures 2–4).

Coprolite ZPAL AbIII/3402 is the biggest specimen (figure 2 and table 1) and contains also the greatest abundance of beetle remains. Remains of beetles of many different species are evidenced by the great disparity of the elytra. Three morphotypes of smaller elytra are found: short and bulky

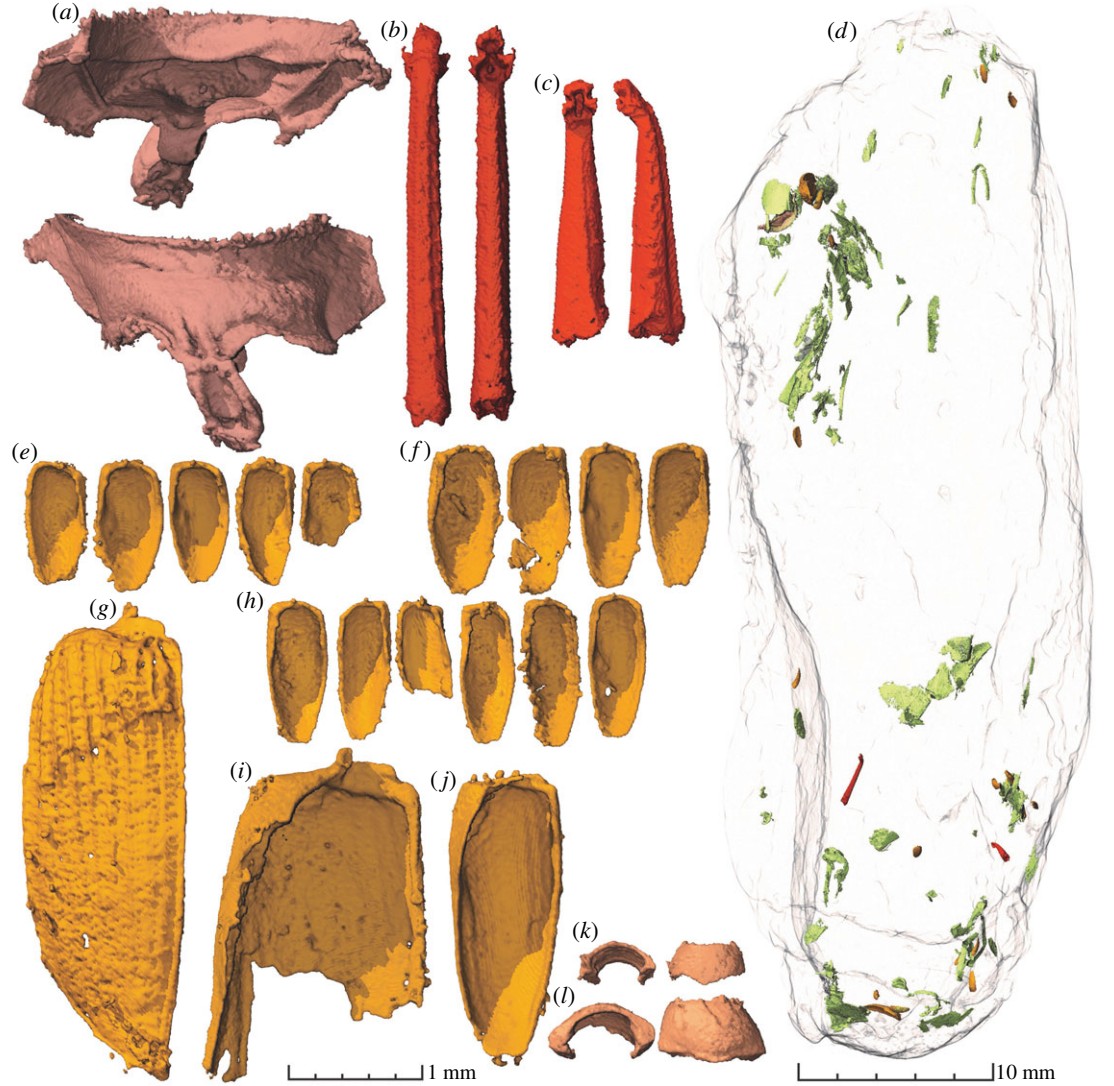

**Figure 2.** Coprolite ZPAL AbIII/3402 and identified inclusions. (*a*) Carabid prosternum. (*b*) Beetle tibia. (*c*) Beetle tibia. (*d*) Entire coprolite in semi-transparent with the identified inclusions as well as some of the indeterminable arthropod/insect remains (green). (*e*) Beetle elytra of morphotype A. (*f*) Beetle elytra of morphotype B. (*g*) Elytron of a polyphagan (?) beetle. (*h*) Beetle elytra of morphotype C. (*i*) Fragmented large elytron. (*j*) Wedge-shaped elytron. (*k,l*) Two beetle pronotums.

specimens (morphotype A); intermediate ones (morphotype B) and elongated specimens (morphotype C). Moreover, three bigger elytra are also found in the coprolite, all differing from one another in ornamentation, shape, size and morphology of the articulatory root (figure 2). The other beetle-bearing coprolites (ZPAL AbIII/3408–3411) contain only the small elytra (figure 3), which are in many cases more poorly preserved than in coprolite ZPAL AbIII/3402. However, the same three morphotypes are encountered in the small coprolite specimens as well. Other enigmatic insect inclusions are also found, although these lack sufficient morphology for proper identification.

Bacterial colonies, represented by densely mineralized (pyrite) and irregular cloud-shaped volumes are found in several of the coprolites (figure 4). In ZPAL AbIII/3402 a large crack spreads out that connects abundant secondary mineralized spheres of similar sizes (gas escape voids; figure 4). Other common features found in the coprolites include secondary mineralized cracks, preserved folds related to the original assembly of the faeces, gas bubbles preserved as voids and secondary mineralized spheres (figure 4). The studied coprolites differ in size and shape from other collected coprolite morphotypes from Krasiejów which were produced by other vertebrates known from the skeletal record in the same site and contain fish remains, bivalves, bone fragments, rare insect remains and/or plant remains (ongoing study). The similar shape, size and contents of the coprolites altogether suggest that they were produced by one animal species. The lack of bones, fish scales or

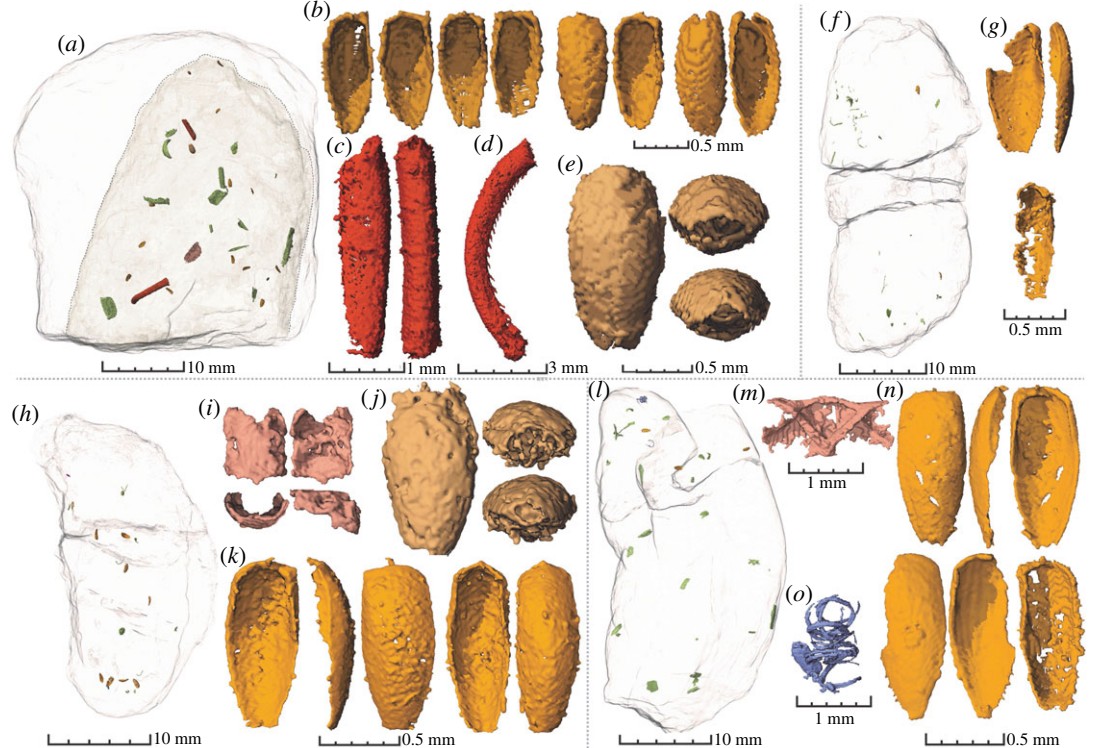

**Figure 3.** Coprolites ZPAL AbIII/3408-3411 with inclusions. ZPAL AbIII/3408: (*a*) Concretion (semi-transparent) with internal fragmentary coprolite with inclusions. (*b*) A selection of six beetle elytra (cf. morphotype B and C in figure 2). (*c*) A part of an insect appendage? (*d*) Enigmatic curved inclusion with denticles on the concave side. (*e*) Abdomen of an unknown arthropod. ZPAL AbIII/3409: (*f*) Semi-transparent coprolite with highlighted inclusions. (*g*) Two beetle elytra. ZPAL AbIII/3410: (*h*) Semi-transparent coprolite with highlighted inclusions. (*i*) Thorax plate of unknown insect. (*j*) Abdomen of an unknown arthropod (same as in *e*). (*k*) Two beetle elytra (cf. morphotype B and C in figure 2). ZPAL AbIII/3411: (*l*) Semi-transparent coprolite with highlighted inclusions. (*m*) A bilateral structure of unknown affinity. (*n*) Three beetle elytra. (*o*) A swirl-shaped inclusion maybe representing some inner insect structure (cf. digestion).

identifiable plant fragments allows us to conclude that the coprolite producers neither had an exclusive carnivorous nor herbivorous diet.

It has been proposed that body size and scat diameter are positively correlated [11,14,26] and considering the size range of the coprolites (maximum diameters 16–22 mm), it is likely that the producer was rather a medium-sized animal than a small animal such as a eucynodont or early lepidosauromorph (cf. modern insectivore mammals).

Studies on modern animals have demonstrated that delicate food remains are underrepresented in faeces, while more resistant objects have the opposite pattern [34]. Thus, we cannot exclude that other food sources such as softer prey and plant fragments, which are not found in the coprolites, formed at least parts of the diet of the coprolite producer. Nevertheless, arthropods (especially insects), and in particular small beetles, were probably the most common prey of this animal judging by their large numbers in the coprolites. Since these remains are very small it implies that the animal either: (1) specifically targeted tiny prey, (2) accidently swallowed the beetles or (3) possessed a structure in the digestive system tract that separated smaller food residues from larger ones, which were regurgitated rather than excreted in the faecal matter. The second hypothesis is weakly supported as beetle remains represent the only identifiable inclusions in all coprolites and that they are rare in coprolites of other morphotypes (ongoing study). The discovery of a few larger beetle remains in the largest coprolites favours the third hypothesis because if the animal possessed such a structure, it would probably grow as the animal became bigger resulting in bigger food remains being able to pass through. Given the fact that the coprolites are of slightly different size, just as the food inclusions, they were probably produced by individuals of slightly different sizes and, perhaps, ontogenetic stages.

In sum, the coprolites of this morphotype were produced by a medium-sized animal that targeted insects as prey. It is likely that animal possessed a system in the digestive tract that separated out

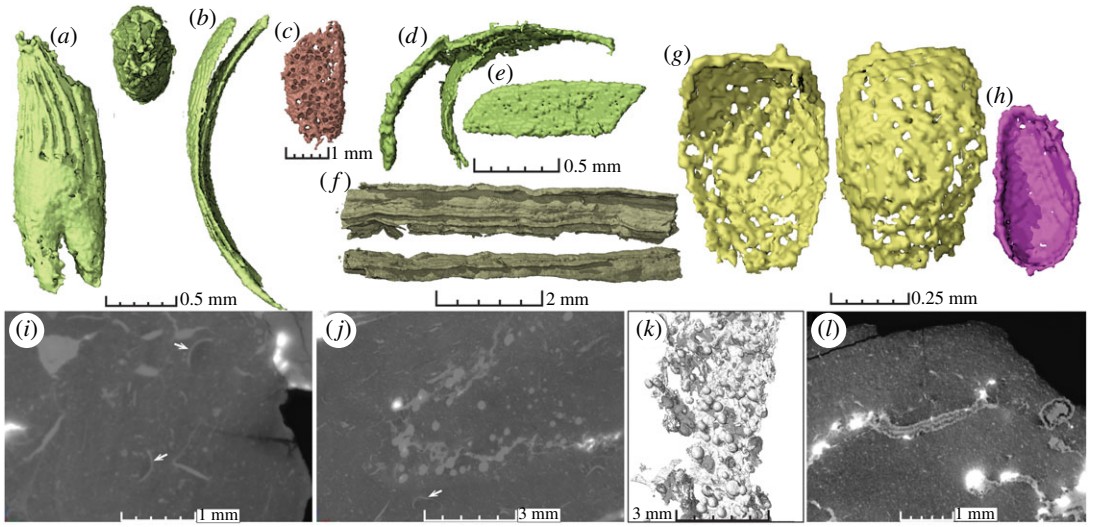

**Figure 4.** Virtual thin sections and other inclusions. (*a*) Enigmatic double-rooted inclusion with striations (in two different views). (*b*) Concave segment of probable insect origin composed of two parts. (*c*) Perforated structure of unknown origin. (*d*) Fragments, perhaps from insect exoskeletons. (*e*) Small 'plate' of unknown origin. (*f*) Elongated structure with ridges. (*g*) A very wide beetle elytron? (*h*) Ostracod carapace (involuntarily ingested?). (*i*) Virtual thin section of coprolite matrix and various insect inclusions. Arrows indicate beetle elytra. (*j*) Virtual thin section with insect remains and mineralized spheres developed around a prominent crack. (*k*) The secondary mineralized spheres and crack in three-dimensional reconstruction. (*l*) Virtual thin section of coprolite matrix with probable bacterial colonies. (*a,b*) ZPAL AbIII/3411; (*c*) ZPAL AbIII/3408; (*d,e*) ZPAL AbIII/3409 (*f,i,j,k*) ZPAL AbIII/3402; (*g,h,l*) ZPAL AbIII/3410.

larger indigestible food remains that were probably regurgitated as pellets instead of passing through the entire digestive system tract (cf. modern birds). These characteristics rule out the majority of the animals known from the body fossil record of the locality as producers (the aetosaur *S. olenkae* was probably an omnivorous/herbivorous scratch digger too large for solely targeting small beetles [35]; the phytosaur and temnospondyls show clear adaptations to piscivory; the large carnivorous rauisuchian *P. silesiacus* was too large and the majority of the other reptiles, as lepidosauromorphs or small archosauromorphs were too small to produce such large faeces) but fit well with the dinosauriform *S. opolensis* (estimated body weight 15 kg), which is known from numerous body fossils in the same fossiliferous interval [30,36]. *Silesaurus opolensis* possesses anatomical characters which are more similar to those of birds rather than other basal dinosaurs. Below follows a list of characters, many of which we interpret as connected to feeding adaptions.

## 3.3. Anatomy and feeding adaptations of *Silesaurus opolensis*

*Silesaurus opolensis* possesses characteristic cranial adaptations that were probably connected to diet [37]. Some of these adaptations are visible in the braincase morphology, which may imply that *S. opolensis* evolved toward a novel feeding behaviour [37]. The new skull reconstruction [38] (and ongoing studies) proposes changes in a number of aspects from previous works [30,36]: the skull was probably shorter and more compact, the antorbital fenestra is reduced compared with other early dinosauriforms, the well-developed jugal has a high and broad contact with the quadratojugal, and the dentary shows two distinct rows of resorption pits. The braincase reconstruction [37] proposes a new arrangement of the paroccipital process, directed ventrally like in birds, reaching the level of the ventral margin of the basioccipital condyle (figure 5). Similar modifications observed in birds have resulted in the dorsoventral expansion of m. complexus (analogous to the 'hatching muscle' in birds) and m. depressor mandibulae, which occupy the dorsolateral part of the posterior aspect of the skull. In adult birds, these muscles support mobility of the head and act strongly on the initial upstroke of the head while drinking [37].

The teeth of the upper and lower jaw are irregularly distributed and oriented laterally with distinct traces of wear [36]. Even though the teeth have a triangular shape, they are blunt (figure 5). The dentine crowns are covered with thin, transparent enamel, which forms longitudinal ridges and grooves. The serrations of both tooth margins are variable but generally not prominent [36]. The bases of the

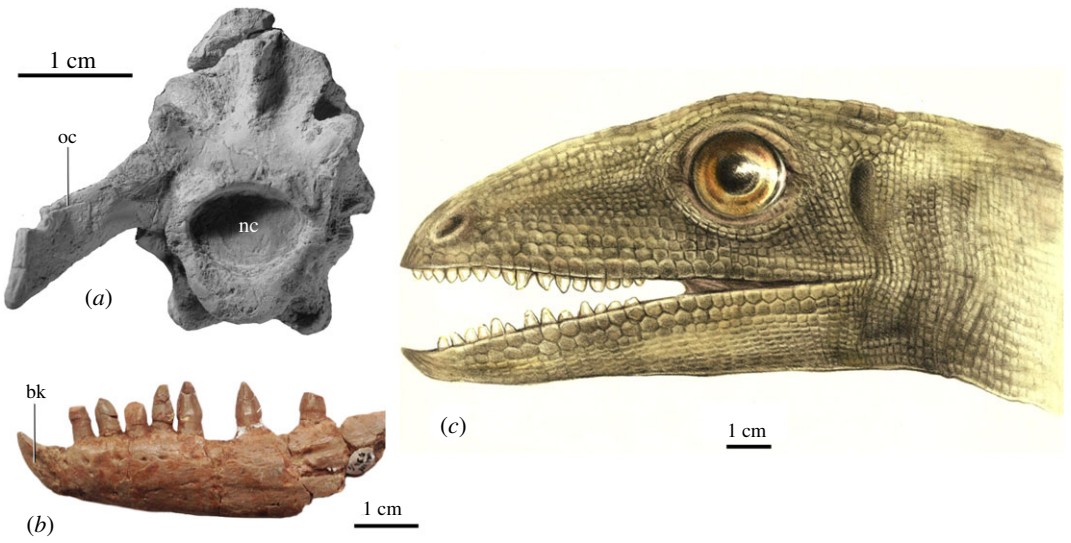

**Figure 5.** The proposed coprolite producer *Silesaurus opolensis* and some anatomical characters. (*a*) Braincase in posterior view (oc, paroccipital process; nc, neural canal). (*b*) Dentary bone in lateral view (bk, beak). (*c*) Life reconstruction of head (based on the skeletal reconstruction of skull presented by Piechowski *et al.* [37]; drawing by Małgorzata Czaja).

crowns are labiolingually expanded, especially on the medial side and often overlapping the adjacent teeth [39]. The teeth are firmly embedded in their alveoli [36] and their number is relatively small. The premaxilla carried five teeth, whereas the maxilla and dentary had 11–12 teeth. Based on its general dental morphology, *S. opolensis* was deduced to be a herbivore [30,36,40]. For example, herbivory on soft objects was inferred from comparisons between the dental microwear patterns of *S. opolensis* and those of extant mammals, although the possibility of omnivory was not confidently ruled out [40]. However, the teeth of *S. opolensis* are neither numerous nor regularly spaced, and they lack the coarse serrations which are typical for herbivores. The orthal jaw movement of *S. opolensis* was much simpler than that of other Late Triassic medium-sized herbivores such as rhynchosaurs, aetosaurs and some therapsids [40]. Therefore, although *S. opolensis* could exploit plant resources, it was most probably not a strict plant-eating archosaur.

The most unusual feature of the *S. opolensis* dentary is its tapering, toothless anterior tip that is hooked upwards (figure 5). Ferigolo & Langer [41] suggested correspondence between this beak-like structure and the predentary of ornithischians, but it differs from that bone because it has a clear mandibular symphysis, and no suture separating the tip from the main part of dentary. Its surfaces are depressed relative to the rest of the mandible and bear indistinct, longitudinal striations and small foramina, suggesting that it was covered with a keratinous beak [36]; an innovative structure found in numerous tetrapod lineages, but especially characteristic for birds where it has an important role during feeding. The premaxilla has a smooth surface and carries teeth to the anterior tip. It is very narrow and the angle between left and right premaxilla was very low. Moreover, the nasal process is mediolaterally thin, which implies a very narrow snout with nostrils directed anteriorly. The premaxilla does not show as obvious evidence for a beak-like structure as the dentary. However, in addition to the sporadic nutrient foramina there are also pores on the anterior part of the premaxilla, which could have provided vascularization to a keratinous cover. Therefore, it is possible that a counterpart to the beak-like structure of the dentary, though not as prominent, was present in the upper jaw.

The discovery of a new specimen of *S. opolensis* with an articulated vertebral column revealed that the neck only consisted of seven vertebrae [42]. This is evidenced by a sudden change in the ribs between the seventh and eighth vertebrae. Delicate and slender ribs are found on all cervicals; the most anterior ones overlapped the next three vertebrae and were more robust than the significantly shorter posteriorly located ribs. The long ribs stiffened the neck, but the construction of the occipital condyle and atlas still enabled a wide range of head movements [42]. More than 400 bones and four partially articulated skeletons of *S. opolensis* have been collected in the upper interval with bones [36]. The bones from upper horizon are preserved in a similar manner, not sorted, and are without any damage. Apart from the bones occurring in the upper horizon,

several bones and single partially articulated skeleton have been found in other bone-bearing beds. The isolated humerus occurred together with the remains of *Polonosuchus silesiacus* and numerous isolated bones and skeleton come from the lower horizon [30]. The studied beetle-bearing coprolites (five specimens) were collected from upper (two specimens) and lower (three specimens) fossiliferous intervals (table 1) and are definitely much rarer in the Krasiejów record than bones of *S. opolensis*.

# 4. Palaeoecological and evolutionary significance

The beak-bearing mandible and the narrow snout [41] could have worked as effective tools for rooting in the litter and pecking insects off the ground, much like modern birds. Large eyes and anteriorly directed nostrils probably participated in food detection (figure 5), and the upward movements could have been supported by the stiffness of the neck. This interpretation fits well with the content of coprolites, which consist of beetles that could well be litter dwellers. Moreover, we suggest that *S. opolensis* had a similar alimentary tract to birds, in which larger food remains were regurgitated, cf. the pellets formed in the gizzard of owls that move upward to the proventriculus and are subsequently regurgitated, and thus never enters the small intestine.

The influence of diets on the early evolution of dinosaur lineage is a subject of ongoing debate [43,44] in which the silesaurids have a key position because they constitute an early group of dinosauriforms or early ornithischians [36,45]. We hope that our results, which imply that *S. opolensis* was probably mostly an insectivorous animal, will spark this discussion and have impact on our understanding of early dinosaur evolution. It should be noted, though, that *S. opolensis* in many regards was a highly specialized animal with several autopomorphies (e.g. see above and [31,36–38]). Also, although not preserved in the coprolites, it is likely that other food sources such as soft prey, plant fragments and larger food items that were regurgitated (so that resistant remains never preserved in the coprolites) constituted at least parts of the animal's diet. It cannot be excluded that beetles were more common during certain periods of the year and represented a seasonal diet for *S. opolensis* during those times. Nevertheless, our results show that coprolites represent an important but largely untapped source of palaeobiological data to unravel the diets of early dinosaurs and their relatives.

# Reconstructions and segmentation

The reconstructions of the scanned data were based on a phase retrieval approach [46,47]. Ring artefacts were corrected using an in-house correction tool [48]. Binned versions (bin2) were calculated to allow faster processing and screening of the samples because the full resolution data were large. The final volumes consist in stacks of 16 bits TIFF images that were converted into JPEG2000 images and subsequently imported and segmented in the software VGStudio MAX version 3.0 (Volume Graphics Inc.).

Data accessibility. The raw data from PPC-SRμCT that support the findings of this study in the form of reconstructed stacks of jpeg2000 images of all coprolites are publicly available in ESRF's paleontological microtomographic database: http://paleo.esrf.fr/picture.php?/2832/category/2226. The studied coprolite specimens are stored in the collection of the Institute of Paleobiology, Polish Academy of Sciences (Warsaw).

Authors' contributions. M.Q., G.N. and P.E.A. designed the project. M.Q. and G.N. performed the scanning. J.V.W. segmented the data from all coprolites except ZPAL AbIII/3402 (segmented by M.Q.), and produced the figures together with M.Q. The part about the anatomy of *Silesaurus* was prepared by R.P., M.T. and G.N. The first version of the manuscript was drafted by M.Q. together with G.N. All authors contributed to the final version.

Competing interests. The authors declare no competing interests.

Funding. This research was supported by Swedish Research Council (grant no. 2017-05248).

Acknowledgements. The coprolites were scanned at the European synchrotron radiation facility, ESRF, in Grenoble (France) as a part of the proposal ES145. Many thanks to Paul Tafforeau for all the help during the scan session and for reconstructing the scan data. We are also thankful to Rolf G. Beutel (Institute of Systematic Zoology and Evolutionary Biology with Phyletic Museum at Friedrich Schiller University Jena, Germany), Evgeny V. Yan (Paleontological Institute of Russian Academy of Sciences, Moscow), and Alexander G. Ponomarenko (Paleontological Institute of Russian Academy of Sciences, Moscow) for helping us with descriptions of some of the beetle remains. Lucas Fiorelli and two anonymous reviewers are acknowledged for their corrections and comments that improved the final version of the paper.

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
