## [Reviewer comments · Royal Society Open Science]

Review History

RSOS-181042.R0 (Original submission)

Review form: Reviewer 1 (Lucas Fiorelli)

Is the manuscript scientifically sound in its present form?

Yes

Are the interpretations and conclusions justified by the results?

Yes

Is the language acceptable?

Yes

Is it clear how to access all supporting data?

Not Applicable

Do you have any ethical concerns with this paper?

No

Have you any concerns about statistical analyses in this paper?

No

Recommendation?

Accept with minor revision (please list in comments)

Comments to the Author(s)

See attached pdf file (Appendix A).

Review form: Reviewer 2 (Andrey G. Sennikov)

Is the manuscript scientifically sound in its present form?

Yes

Are the interpretations and conclusions justified by the results?

Yes

Is the language acceptable?

Yes

Is it clear how to access all supporting data?

Yes

Do you have any ethical concerns with this paper?

No

Have you any concerns about statistical analyses in this paper?

Yes

Recommendation?

Accept as is

Comments to the Author(s)

It's is very interesting paper about the possible diet of a Late Triassic dinosauriform *Silesaurus opolensis*. Such conclusion based both on the data from coprolites and on morphology of *Silesaurus*. The insectivore for dinosaur precursors is very probable hypothesis.

Review form: Reviewer 3 (Michael Benton)

Is the manuscript scientifically sound in its present form?

Yes

Are the interpretations and conclusions justified by the results?

Yes

Is the language acceptable?

Yes

Is it clear how to access all supporting data?

Yes

Do you have any ethical concerns with this paper?

No

Have you any concerns about statistical analyses in this paper?

No

Recommendation?

Accept with minor revision (please list in comments)

Comments to the Author(s)

This is a remarkable discovery and the evidence is carefully considered. It appears to show conclusive evidence that *Silesaurus*, a close relative of the first dinosaurs ate beetles, and that it selected particular types of beetle. Hitherto, from its anatomy most palaeontologists would have assumed it targeted small tetrapods as its main food. The evidence comes from multiple examples of coprolites which the authors have examined using state-of-the-art methods. They make convincing arguments that the coprolites come from *Silesaurus*, and so make the link.

Pages 5-6: In assigning the coprolites to *Silesaurus*, say more about the relative distributions of coprolites and bones of this animal between the two fossiliferous level in which both were found. Also, say more about proportions of the skeletal taxa and proportions of the coprolite types – small sample sizes, I know, but this might be informative.

Page 7: I'm not sure about the statement that the supposedly bird-like braincase implies *Silesaurus* had 'bird-like feeding behaviour', Unless you can make a clear and plausible causal connection between the two – i.e. which aspect of the braincase is actually bird-like and is there published data to show this uniquely houses a part of the brain associated with whatever you mean by 'bird-like feeding behaviour'. Otherwise, drop all this and the figure – I think it's all irrelevant (Page 7, lines 11-32).

Page 7, line 32: You flip from braincase to teeth in the same paragraph; move the tooth stuff to the material about *Silesaurus* feeding – former views, current views, evidence in the teeth and jaws that they ate beetles.

Page 9, line 11: I've never heard Silesauridae were paraphyletic – all the recent, authoritative cladistic analyses make them a clade and their relationships are clear – e.g. papers by Nesbitt, Irmis, etc. Omit this or explain why (with evidence) you reject the recent phylogenetic papers.

Maybe consider whether this could be a seasonal diet for *Silesaurus* – feeding on beetles when they are abundant, and coprolites all date from the same season of the year? Yes or no?

4/42: at the Krasiejów locality

5/19: abundancy = abundance

5/43: (figure 4). The [insert space]

6/11: 34 = [34]

6/30: such structure = such a structure

7/19: dinosauriformes = dinosauriforms

8/6: of *Silesaurus* dentary = of the *Silesaurus* dentary

8/40: 40 = [40]

Decision letter (RSOS-181042.R0)

08-Nov-2018

Dear Dr Niedźwiedzki

On behalf of the Editors, I am pleased to inform you that your Manuscript RSOS-181042 entitled "Beetle-bearing coprolites possibly reveal the diet of a Late Triassic dinosauriform" has been accepted for publication in Royal Society Open Science subject to minor revision in accordance with the referee suggestions. Please find the referees' comments at the end of this email.

The reviewers and handling editors have recommended publication, but also suggest some minor revisions to your manuscript. Therefore, I invite you to respond to the comments and revise your manuscript.

- Ethics statement

- Data accessibility

<http://datadryad.org/submit?journalID=RSOS&manu=RSOS-181042>

- Competing interests

- Authors' contributions

AB carried out the molecular lab work, participated in data analysis, carried out sequence alignments, participated in the design of the study and drafted the manuscript; CD carried out

the statistical analyses; EF collected field data; GH conceived of the study, designed the study, coordinated the study and helped draft the manuscript. All authors gave final approval for publication.

- Acknowledgements

- Funding statement

Because the schedule for publication is very tight, it is a condition of publication that you submit the revised version of your manuscript before 17-Nov-2018. Please note that the revision deadline will expire at 00.00am on this date. If you do not think you will be able to meet this date please let me know immediately.

- 1) A text file of the manuscript (tex, txt, rtf, docx or doc), references, tables (including captions) and figure captions. Do not upload a PDF as your "Main Document";
- 2) A separate electronic file of each figure (EPS or print-quality PDF preferred (either format should be produced directly from original creation package), or original software format);
- 3) Included a 100 word media summary of your paper when requested at submission. Please ensure you have entered correct contact details (email, institution and telephone) in your user account;
- 4) Included the raw data to support the claims made in your paper. You can either include your data as electronic supplementary material or upload to a repository and include the relevant doi within your manuscript. Make sure it is clear in your data accessibility statement how the data can be accessed;
- 5) All supplementary materials accompanying an accepted article will be treated as in their final form. Note that the Royal Society will neither edit nor typeset supplementary material and it will

be hosted as provided. Please ensure that the supplementary material includes the paper details where possible (authors, article title, journal name).

on behalf of Dr Julia Brenda Desojo (Associate Editor) and Prof. Jon Blundy (Subject Editor)
openscience@royalsociety.org

Associate Editor Comments to Author (Dr Julia Brenda Desojo):

Dear Authors, please read and incorporate the reviewer comments and suggestions, specialize R1 about :The main problem is the results justify the interpretations and conclusions, but could also justify other interpretations, such as the coprolite producer could be another species of archosaur (case discarded by the authors. Following this idea, I strongly recommend the authors give more support to discard the aetosaur *Stagonolepis olenkae* as a coprolite producer (e.g. similar body size that *S. opolensis*, insectivorous habits suggesting habits -see the recent paper published about *S. olenkae* and the reference about it

- Drózdź (2018), Osteology of a forelimb of an aetosaur *Stagonolepis olenkae* (Archosauria: Pseudosuchia:Aetosauria) from the Krasiejów locality in Poland and its probable adaptations for a scratch-digging behavior. PeerJ 6:e5595; DOI 10.7717/peerj.5595

Reviewer comments to Author:

Reviewer: 1

Comments to the Author(s)

See attached pdf file

Reviewer: 2

Comments to the Author(s)

It's is very interesting paper about the possible diet of a Late Triassic dinosauriform *Silesaurus opolensis*. Such conclusion based both on the data from coprolites and on morphology of *Silesaurus*. The insectivore for dinosaur precursors is very probable hypothesis.

Reviewer: 3

Comments to the Author(s)

This is a remarkable discovery and the evidence is carefully considered. It appears to show conclusive evidence that *Silesaurus*, a close relative of the first dinosaurs ate beetles, and that it selected particular types of beetle. Hitherto, from its anatomy most palaeontologists would have assumed it targeted small tetrapods as its main food. The evidence comes from multiple examples of coprolites which the authors have examined using state-of-the-art methods. They make convincing arguments that the coprolites come from *Silesaurus*, and so make the link.

Pages 5-6: In assigning the coprolites to *Silesaurus*, say more about the relative distributions of coprolites and bones of this animal between the two fossiliferous level in which both were found. Also, say more about proportions of the skeletal taxa and proportions of the coprolite types – small sample sizes, I know, but this might be informative.

Page 7: I'm not sure about the statement that the supposedly bird-like braincase implies *Silesaurus* had 'bird-like feeding behaviour', Unless you can make a clear and plausible causal connection between the two – i.e. which aspect of the braincase is actually bird-like and is there published data to show this uniquely houses a part of the brain associated with whatever you mean by 'bird-like feeding behaviour'. Otherwise, drop all this and the figure – I think it's all irrelevant (Page 7, lines 11-32).

Page 7, line 32: You flip from braincase to teeth in the same paragraph; move the tooth stuff to the material about *Silesaurus* feeding – former views, current views, evidence in the teeth and jaws that they ate beetles.

Page 9, line 11: I've never heard *Silesauridae* were paraphyletic – all the recent, authoritative cladistic analyses make them a clade and their relationships are clear – e.g. papers by Nesbitt, Irmis, etc. Omit this or explain why (with evidence) you reject the recent phylogenetic papers.

Maybe consider whether this could be a seasonal diet for *Silesaurus* – feeding on beetles when they are abundant, and coprolites all date from the same season of the year? Yes or no?

4/42: at the Krasiejów locality

5/19: abundancy = abundance

5/43: (figure 4). The [insert space]

6/11: 34 = [34]

6/30: such structure = such a structure

7/19: dinosauriformes = dinosauriforms

8/6: of *Silesaurus* dentary = of the *Silesaurus* dentary

8/40: 40 = [40]

Author's Response to Decision Letter for (RSOS-181042.R0)

See Appendix B.

RSOS-181042.R1 (Revision)

Review form: Reviewer 1 (Lucas Fiorelli)

Is the manuscript scientifically sound in its present form?

Yes

Are the interpretations and conclusions justified by the results?

Yes

Is the language acceptable?

Yes

Is it clear how to access all supporting data?

Yes

Do you have any ethical concerns with this paper?

No

Have you any concerns about statistical analyses in this paper?

No

Recommendation?

Accept as is

Comments to the Author(s)

No more comments

Review form: Reviewer 3 (Michael Benton)

Is the manuscript scientifically sound in its present form?

Yes

Are the interpretations and conclusions justified by the results?

Yes

Is the language acceptable?

Yes

Is it clear how to access all supporting data?

Yes

Do you have any ethical concerns with this paper?

No

Have you any concerns about statistical analyses in this paper?

No

Recommendation?

Accept as is

Comments to the Author(s)

None

Decision letter (RSOS-181042.R1)

12-Feb-2019

Dear Dr Niedźwiedzki,

I am pleased to inform you that your manuscript entitled "Beetle-bearing coprolites possibly reveal the diet of a Late Triassic dinosauriform" is now accepted for publication in Royal Society Open Science.

on behalf of Dr Julia Brenda Desojo (Associate Editor) and Professor Jon Blundy (Subject Editor)
openscience@royalsociety.org

Reviewer comments to Author:

Reviewer: 3

Comments to the Author(s)

None

Reviewer: 1

Comments to the Author(s)

No more comments

Appendix A

Revision of the Royal Society Open Science manuscript ID RSOS-181042

Title: *Beetle-bearing coprolites possibly reveal the diet of a Late Triassic dinosauriform* (by Qvarnström et al.).

The manuscript of Martin Qvarnström and col. is suitable for publication in **Royal Society Open Science** and will have high impact with important paleobiological implications.

The manuscript is apparently well written although honesty I do not feel empowered to perform the English language due to I am not an Anglophone.

The paper is a very solid piece of work, in line with recent papers published by them. This shows precisely how the developed of coprology is essential for the knowledge of palaeobiology, palaeophysiology, palaeocommunities, and ancient ecosystems. Although some of the information has been published recently (Qvarnström et al. 2017 - DOI:10.1038/s41598-017-02893-9), I think it is extremely necessary beforehand to develop these kind of coprology studies.

Despite this, I can do some comments and improve as well some aspects of the manuscript.

General remarks:

Scientifically the manuscript is solid, well-structured and organized, although the objectives are not clearly stated at the end of the introductory section.

Is well written in a clear and concise way and having a well development across the text. The title is well, reflecting clearly the work content, and the study, its findings and conclusions are clear in the abstract. However, the rest of the manuscript presents some minor problems to be considered and discussed; for that reason, I made just some comments and revisions that must be addressed in the main text before its publication.

Specific comments

- At the end of the introductory section, the objectives are not clearly stated.
- It necessary to justify more accurately why the coprolite belonged to the dinosauriform *Silesaurus* and not, for example, to the aetosaur *Stagonolepis olenkae*. Recently, Drózd (2018 - DOI 10.7717/peerj.5595) suggested that this aetosaur could have been omnivore, specialized in coleopterans, and having produced the same coprolite (and was found in the same levels).
- So, this is directly linked to the conclusions and implications about the suggested diet for dinosauriformes. This is the main problem, because the results justify the interpretations and conclusions, but could also justify other interpretations, such as the coprolite producer could be another species of archosaur (case discarded). In this sense, the authors must support their conclusions more strongly.
- I suggest including a supplementary with videos and interactive 3D views of the coprolites.

Page 1, lines 30-31: ...however, also CAN reflect the...

Page 4, lines 23-24: The terrestrial community was composed of small reptiles... Could you be more precise in that assignment?

Page 6, line 11: check reference format.

Page 6, line 45: this is not in tune with the suggestions of Drózd (2018), and it is precisely the weakness of the manuscript.

Note on this regard: I think too that the precursors of dinosaurs (e.g., dinosauriformes) would have been omnivorous and their diet constituted mostly by arthropods; in fact I think that the Triassic trophic chains initially were supported by arthropods (and plants, of course), but I observe that there is an equal support (*Silesaurus* / *Stagonolepis*) towards the producer of the coprolites.

Page 8, line 48: check reference format.

Final comment:

Again, and so to summarize, I like to point out that the manuscript is very interesting and the science here is great. Qvarnström et al.'s manuscript is very reasonable and I look forward to seeing it published in Royal Society Open Science but just after some key and necessary revisions.

Sincerely,

Dr. Lucas Fiorelli

CRILAR-CONICET

La Rioja, Argentina. October 4, 2018

Appendix B

Reviewer comments to Author:

Reviewer: 1

The manuscript of Martin Qvarnström and col. is suitable for publication in Royal Society Open Science and will have high impact with important paleobiological implications. The manuscript is apparently well written although honesty I do not feel empowered to perform the English language due to I am not an Anglophone.

The paper is a very solid piece of work, in line with recent papers published by them. This shows precisely how the developed of coprology is essential for the knowledge of palaeobiology, palaeophysiology, palaeocommunities, and ancient ecosystems. Although some of the information has been published recently (Qvarnström et al. 2017 - DOI:10.1038/s41598-017-02893-9), I think it is extremely necessary beforehand to develop these kind of coprology studies. Despite this, I can do some comments and improve as well some aspects of the manuscript.

General remarks: Scientifically the manuscript is solid, well-structured and organized, although the objectives are not clearly stated at the end of the introductory section. Is well written in a clear and concise way and having a well development across the text. The title is well, reflecting clearly the work content, and the study, its findings and conclusions are clear in the abstract. However, the rest of the manuscript presents some minor problems to be considered and discussed; for that reason, I made just some comments and revisions that must be addressed in the main text before its publication.

Specific comments

- At the end of the introductory section, the objectives are not clearly stated.

This is now added.

- It necessary to justify more accurately why the coprolite belonged to the dinosauriform Silesaurus and not, for example, to the aetosaur *Stagonolepis olenkae*. Recently, Drózd (2018 - DOI 10.7717/peerj.5595) suggested that this aetosaur could have been omnivore, specialized in coleopterans, and having produced the same coprolite (and was found in the same levels).

Drózd (2018) analyzed the forelimbs of *Stagonolepis* and inferred probable adaptations to scratch digging. In the same paper, Drózd (2018) discussed the diet of recent scratch-diggers and concluded that the large size of *Stagonolepis* suggest a diet/feeding strategy more similar to wild bores than to such, much smaller, insectivorous scratch diggers. We believe too, that *Stagonolepis* was probably too big to have produced these small coprolites, and specifically targeted such small

beetles (we added this more clearly in the manuscript now, and referenced this paper). More likely, it had a diverse menu.

- So, this is directly linked to the conclusions and implications about the suggested diet for dinosauriformes. This is the main problem, because the results justify the interpretations and conclusions, but could also justify other interpretations, such as the coprolite producer could be another species of archosaur (case discarded). In this sense, the authors must support their conclusions more strongly.

We have in the coprolite collection from Krasiejów specimens which probably represent fossilized dungs of *Stagonolepis*. More on this topic will be in the next publication (in prep.) with a description of all specimens from the site.

- I suggest including a supplementary with videos and interactive 3D views of the coprolites.

Page 1, lines 30-31: ...however, also CAN reflect the...

added

Page 4, lines 23-24: The terrestrial community was composed of small reptiles...
Could you be more precise in that assignment?

Examples are now given

Page 6, line 11: check reference format.

corrected

Page 6, line 45: this is not in tune with the suggestions of Drózd (2018), and it is precisely the weakness of the manuscript.

See comment above

Note on this regard: I think too that the precursors of dinosaurs (e.g., dinosauriformes) would have been omnivorous and their diet constituted mostly by arthropods; in fact I think that the Triassic trophic chains initially were supported by arthropods (and plants, of course), but I observe that there is an equal support (*Silesaurus* / *Stagonolepis*) towards the producer of the coprolites.

Page 8, line 48: check reference format.

corrected

Final comment: Again, and so to summarize, I like to point out that the manuscript is very interesting and the science here is great. Qvarnström et al.'s manuscript is very

reasonable and I look forward to seeing it published in Royal Society Open Science but just after some key and necessary revisions.

Sincerely, Dr. Lucas Fiorelli CRILAR-CONICET La Rioja, Argentina.

Reviewer: 2

Comments to the Author(s)

It's a very interesting paper about the possible diet of a Late Triassic dinosauriform *Silesaurus opolensis*. Such conclusion based both on the data from coprolites and on morphology of *Silesaurus*. The insectivore for dinosaur precursors is a very probable hypothesis.

Reviewer: 3

Comments to the Author(s)

This is a remarkable discovery and the evidence is carefully considered. It appears to show conclusive evidence that *Silesaurus*, a close relative of the first dinosaurs ate beetles, and that it selected particular types of beetle. Hitherto, from its anatomy most palaeontologists would have assumed it targeted small tetrapods as its main food. The evidence comes from multiple examples of coprolites which the authors have examined using state-of-the-art methods. They make convincing arguments that the coprolites come from *Silesaurus*, and so make the link.

Pages 5-6: In assigning the coprolites to *Silesaurus*, say more about the relative distributions of coprolites and bones of this animal between the two fossiliferous level in which both were found. Also, say more about proportions of the skeletal taxa and proportions of the coprolite types – small sample sizes, I know, but this might be informative.

Supplemented.

Page 7: I'm not sure about the statement that the supposedly bird-like braincase implies *Silesaurus* had 'bird-like feeding behaviour', Unless you can make a clear and plausible causal connection between the two – i.e. which aspect of the braincase is actually bird-like and is there published data to show this uniquely houses a part of the brain associated with whatever you mean by 'bird-like feeding behaviour'. Otherwise, drop all this and the figure – I think it's all irrelevant (Page 7, lines 11-32).

Corrected and improved.

Page 7, line 32: You flip from braincase to teeth in the same paragraph; move the tooth stuff to the material about *Silesaurus* feeding – former views, current views, evidence in the teeth and jaws that they ate beetles.

This is now separated into two different paragraphs.

Page 9, line 11: I've never heard Silesauridae were paraphyletic – all the recent, authoritative cladistic analyses make them a clade and their relationships are clear – e.g. papers by Nesbitt, Irmis, etc. Omit this or explain why (with evidence) you reject the recent phylogenetic papers.

removed

Maybe consider whether this could be a seasonal diet for Silesaurus – feeding on beetles when they are abundant, and coprolites all date from the same season of the year? Yes or no?

We have no data from the coprolites that suggest they were all produced during the same season. Nevertheless, we cannot exclude that this beetle-dominated diet was seasonal (in the same way we cannot exclude dietary components that simply were not preserved/entering the coprolites). This is a good suggestion that is now incorporated in the manuscript.

4/42: at the Krasiejów locality

corrected

5/19: abundancy = abundance

corrected

5/43: (figure 4). The [insert space]

corrected

6/11: 34 = [34]

corrected

6/30: such structure = such a structure

corrected

7/19: dinosauriformes = dinosauriforms

corrected

8/6: of Silesaurus dentary = of the Silesaurus dentary

corrected

8/40: 40 = [40]

corrected